# Aerosol size determination via light scattering of viruses and protein complexes
Lena Worbs[1,3], Tej Varma Yenupuri[1,3], Tong You[1] & Filipe R. N. C. Maia [1,2] ✉

The study of ultrafine particle aerosols, those with particle diameters of 100 nm or less, is important due to their impact on our health and environment. However, given their small sizes, such particles can be difficult to measure and trace. Most common optical methods are unable to reach this size range. Other methods exist but incur other limitations, such as the need for electrically charged particles. Here we show how light scattering can be used to detect and measure the size and location of single viruses and protein complexes forming an aerosol beam, as well as trace their path. We were able to detect individual particles down to 16 nm in diameter. The primary purpose of our instrument is to monitor the delivery of single bioparticles to the focus of an X-ray laser to image those particles, but it has the potential to study any other aerosols such as those resulting from ultrafine sea spray, with important consequences for cloud formation and climate modeling, or from combustion, responsible for most air pollution and resulting health impacts.

Small particles have a big impact on our life. The range of particles with a diameter below 100 nm, also known as ultrafine particles or PM0.1, includes many important biological entities such as protein complexes and most viruses. The recent COVID-19 pandemic made it evident that understanding the behavior of aerosols containing these minuscule particles holds great importance[1,2]. Furthermore, ultrafine particle aerosols are an important contributor to air pollution, mainly derived from combustion processes, such as in road vehicles. This air pollution inflicts substantial societal and economic burdens on the global environment[3]. Ultrafine particles directly impact our health[4], first by attacking the respiratory system[5,6], like other larger particles, but then spreading to other parts of the body[7]. They are involved in cardiovascular diseases[8,9], damage to the central nervous system[10,11], increased diabetes[12] and cancer rates[13]. Naturally generated ultrafine particles, such as the one from sea spray, also have an important impact on our climate through their role in cloud formation[14,15].

Ultrafine particle aerosols are difficult to measure and visualize due to their small size. Differential mobility particle sizers (DMPS) and scanning mobility particle sizers (SMPS), a combination of a differential mobility analyzer[16] (DMA) with a condensation particle counter (CPC), are probably the most common way to study such aerosols, measuring their electric mobility. However they have limited temporal resolution and only work with charged aerosols. Optical methods, such as optical particle counters[17] and aerodynamic particle sizers[18], while cheap, are typically limited to particles above 100 nanometers in diameter[19]. Microresonator-based

systems have shown promise in measuring nanoparticles[20], but they require the deposition of the aerosol on a substrate and are unable to resolve the spatial distribution of the particles. Recently the ability of mass photometry[21] to measure the mass of individual, unlabeled biomolecules in dilute solution to high accuracy, using light scattering, has opened a host of exciting applications[22,23]. This inspired us to see how far we could push the direct imaging of PM0.1 aerosols by laser scattering.

Aerosols are also used to carry biological particles to be studied in native mass spectrometry[24] and single-particle X-ray diffractive imaging (SPI)[25] on various samples ranging in size from viruses down to single proteins[26–28]. In SPI, 2D X-ray diffraction patterns are recorded from randomly oriented non-crystalline isolated (bio-)nanoparticles to reconstruct their 3D electron density resulting in their structure. Using short X-ray pulses, a diffraction pattern of an intact particle can be recorded before its destruction[29].

In SPI an aerosol is generated from the sample solution using electrospray ionisation (ESI)[30,31]. The charged aerosol is then neutralized and aerodynamically focused into a particle beam that is intersected by an X-ray pulse in a vacuum chamber[32–34].

To monitor the quality of the particle beam delivered to the X-rays, the aerosol droplet distribution is typically measured after the ESI source using a differential mobility analyser[16] in combination with a condensation particle counter (DMA-CPC). However, this distribution does not necessarily reflect the size distribution in the interaction region e.g., due to transmission

[1]Department of Cell and Molecular Biology, Laboratory of Molecular Biophysics, Uppsala University, Uppsala, Sweden. [2]NERSC, Lawrence Berkeley National Laboratory, Berkeley, CA, USA. [3]These authors contributed equally: Lena Worbs, Tej Varma Yenupuri. ✉e-mail: filipe.maia@icm.uu.se

discrepancies of different sizes in a heterogeneous sample and differences in droplet evaporation.

Previous nanoparticle sizing methods in particle beams were performed using optical scattering[32,35,36], but were restricted to sizes larger than 40 nm.

In this work, we extend the use of optical scattering microscopy to size small viruses and biological macromolecular complexes in vacuum, down to 16 nm in diameter, which was previously out of reach due to their small size.

By improving the focus of the incident laser and optimizing the experiment geometry we have greatly increased the range of observable particle sizes at the cost of a decreased field of view. This makes it possible to measure the particle size distribution of biologically relevant samples directly in the interaction region. Our method has the potential to make sample delivery of isolated protein complexes in SPI experiments[28] simpler and more robust opening the way for the study of single-particle ultrafast dynamics. This method can also be used as a general aerosol sizing technique when standard approaches are impractical or impossible.

## Methods
### Experimental setup
To size the nanoparticles and characterize the particle-beam properties, we modified our previously used Rayleigh-microscopy setup[32]. In the new design, the laser beam, the particle beam and the CMOS camera are perpendicular to each other (see Fig. 1). The green pulsed Nd:YAG laser (Quantel Evergreen 25100, $\lambda = 532$ nm, pulse duration 10 ns, pulse energy 93.3 mJ, repetition rate 15 Hz, single pulse mode) beam is focused using a 50 mm plano-convex lens onto the particle beam, creating a focus size of 35 µm (FWHM) that illuminates the nanoparticles which are imaged through a custom microscope system. The camera and tube lens are set up outside the experimental chamber, whereas the objective is placed inside. The light scattered by the illuminated particles passes through an infinity-

corrected 10x (NA = 0.45, 2.7 µm depth of field) objective lens, a 200 mm tube lens and is captured by a CMOS camera (Hamamatsu Orca Flash 4.0 V2).

As a comparison measurement we used an electrostatic classifier (TSI model 3080), a differential-mobility analyzer (TSI model 3081) and a condensation particle counter (TSI model 3786) (DMA-CPC) to measure the aerodynamic diameters of the nanoparticles.

The aerosol was generated by an ESI source as previously described[31]. The aerosol then goes through two skimmer pairs to compensate for the high gas input from an ESI source and is focused onto the interaction region by an aerodynamic lens stack (TSI AFL-100). In the skimmer pairs the excess gas is skimmed away using one scroll pump per skimmer box. The particles enter the aerodynamic lens with an entrance pressure of 0.56 mbar and exit through a 1.5 mm aperture, 2 mm above the interaction region in the experimental chamber which is kept at $6 \times 10^{-5}$ mbar. The schematic setup is shown in the Supplementary Fig. 3.

We studied three biological samples with diameters ranging from 13 to 27 nm and for calibration polystyrene spheres (PS) with diameters ranging from 18 to 59 nm. All samples were suspended in 20 mM ammonium acetate (AmAc) solution. We used Bacteriophage MS2, ribosomes (70S ribosomes isolated from *Escherichia coli*) and apo-ferritin (from equine spleen), with measured aerodynamic diameters of 25.9 nm, 19.5 nm, and 13.6 nm, respectively. From here on, we refer to the biosamples as MS2, ribosomes, and ferritin, respectively. The sample details including manufacturer's size, DMA-CPC size distributions and particle solution concentrations are given in Supplementary Note 1.

### Data collection and analysis
We recorded frames to measure the scattering intensity of the different particles with the laser in single-pulse mode. We collected multiple datasets with a minimum of 1000 frames, where each frame corresponds to a single laser shot. The number of frames was increased to 2000 and 3000 frames for particles with smaller diameters due to decreased hit rates. The camera frames were analyzed with our open-source software package (https://github.com/Toonggg/spts) to determine the particle position and scattered intensity. An example of a single raw frame containing a particle hit is shown in Fig. 2a. The raw data frames were corrected for dead pixels on the camera (see center of frame in Fig. 2a) and the mean laser background, which was recorded without sample being delivered, was subtracted as shown in Fig. 2b. The images were then denoised by convolving them with a Gaussian kernel with a standard deviation of 1.6 pixels. We then threshold the denoised image to determine the pixels with photons with a fixed threshold of 20. Particle hits were determined from the thresholded data and the center was defined as the pixel with the maximum intensity. The total scattered signal per hit was determined by integrating all pixels within a circular window large enough to account for the point spread function of the system. We limited the analysis to focused particles in the laser illumination region.

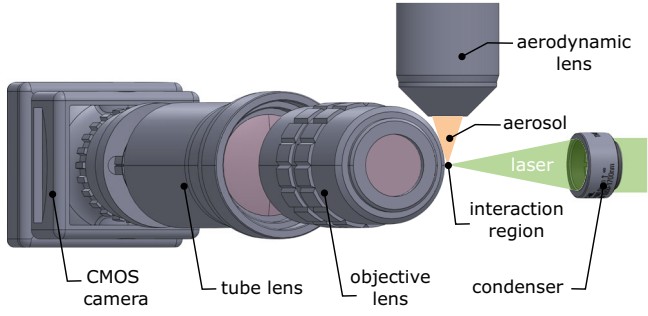

**Fig. 1 | Experimental Rayleigh-scattering microscopy setup.** The aerodynamic lens, the objective lens and the optical beam path are perpendicular to each other.

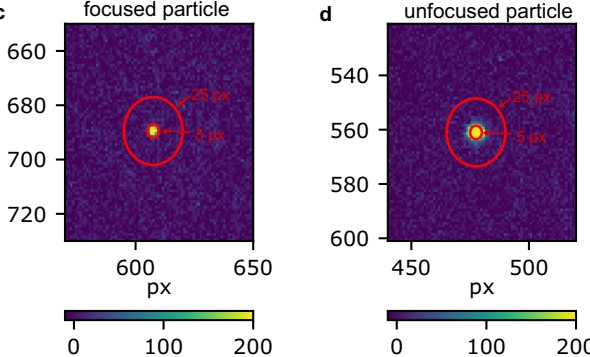

**Fig. 2 | Camera frame examples. a** Raw camera image. **b** Background subtracted camera image in the region of interest with a particle hit. **c** The particle hit from b.) with the two circular windows used for analysis. A focused particle is shown. Focused

particle examples for all particle sized used are shown in Supplementary Fig. 4. **d** An example of an unfocused particle hit.

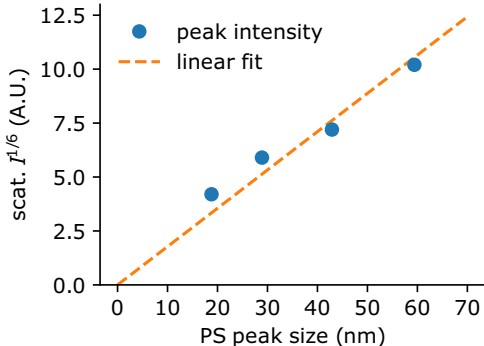

**Fig. 3 | Calibration curve.** Measured peak scattering intensity depending on the PS particle peak diameter with a linear fit (dashed line).

To select the focused particles, we ran the data analysis twice: first with a large circular window of 25 px diameter and a second time with a 5 px circular window. We defined focused particles as those where more than 90% of the scattering intensity determined in the large window fell inside the small 5 px window. Examples of a focused and an unfocused particle hit are shown in Fig. 2c, d, respectively. For each particle size, we determined the peak scattering intensity of all the selected particles.

To trace individual particles and determine the particle's speed, we recorded frames with the laser in double-pulse mode. The delay of the pulses was set to 0.3 μs for the PS sample and to 0.2 μs for the biosamples. Each frame corresponds to a double laser shot. The data analysis was performed similar to the data analysis of the single pulse data with the extension of finding frames with a double hit and determining the difference in distance between the hits to determine the particle's speed.

## Results and discussion

We calibrated the apparatus by measuring the scattering intensity from mono-disperse aerosols, each containing PS in the size range 18 to 59 nm (peak diameter according to DMA-CPC measurements, see Supplementary Fig. 1a). For particles much smaller than the wavelength of the illumination ($d_p \ll \lambda_{\mathrm{laser}}$), the scattering follows Rayleigh's law so the scattered intensity is proportional to the sixth power of their radius. The measured peak sixth root intensity is shown in Fig. 3 as a function of the measured particle peak diameter. We used `scipy.optimize.curve_fit` to fit a linear function through the $\sqrt[6]{I(d_p)}$ data (dashed line). The slope of the linear fit is 0.178 with a root mean squared error of 0.009.

The retrieved size histograms calculated from the scattering data of the PS samples based on the calibration is shown in Supplementary Fig. 5 in comparison to the DMA-CPC size histogram.

We determined the lower detection limit in our setup to be 16 nm. This value is a result of the shot-to-shot pixel intensity difference (noise) and the background level (background). The noise distribution is shown in Fig. 4a. The dashed black line shows the average distribution. In Fig. 4b, we show the PS hit intensity to background ratio (blue) with a sixth-root fit through the data (orange line). For the large 59 nm PS particles, the signal is 240 times larger than the background and this ratio decreases to 3.3 for the 18 nm PS particles. The lower detection limit is determined by the shot-to-shot noise. This is represented by the gray line. The value is taken as the $1/e^2$ value of the distribution in Fig. 4a. The two lines cross at a particle size of 16 nm, which we set as our lower detection limit. For this size, the intensity would only be 2.8 times larger than the background.

We recorded scattering data sets from three biological samples and calculated their size histogram based on the scattering-size calibration parameter we obtained from the polystyrene calibration data. The resulting particle size histogram is shown in blue in Fig. 5. We compare this distribution with the measured size histogram from a DMA-CPC measurement (orange). The scattering and DMA-CPC size histograms of MS2 show

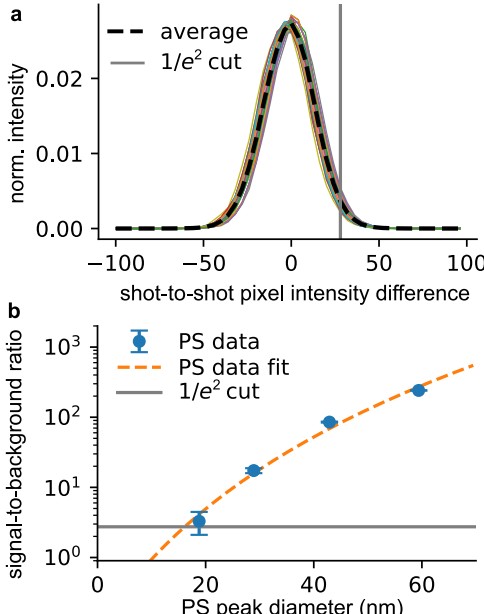

**Fig. 4 | Background estimation. a** Normalized shot-to-shot pixel background noise histogram (colored lines) with the average distribution (black line). **b** Signal-to-background ratio of the PS calibration data depending on the PS size (blue) with a fit to the data (dashed orange line). The gray line represents the $1/e^2$ value of the shot-to-shot noise from the distribution shown in (**a**). The error bar is given as one standard deviation.

great similarity, see Fig. 5a. Besides a large peak at small particle diameters, $d_p < 20$ nm, a separate peak is observed. The DMA-CPC distribution (orange) shows the MS2 particle diameter at 26 nm. The particle diameter from the scattering data has a wider range (around 24 nm to 26 nm). The smaller diameters could occur due to the extra drying and partial collapse of the viral capsids when exposed to vacuum conditions, as has been previously observed[37,38], and showcases the need for a diagnostic that occurs in the same conditions as the real experiment. A large number of smaller particles with $d_p < 20$ nm are visible in both the scattering and the DMA-CPC data. We assume the small particles to be fragments of the virus due to long storage times and insufficient buffer exchange and cleaning of the sample solution. Initial MS2-empty droplets in the electrospray process can contain aggregates of the buffer residues and create spheres in this size range that we refer to as buffer balls. To support this idea, we have recorded control data while injecting the MS2 buffer without MS2 virus particles. We observed particle hits with small diameters only. This data is presented in Supplementary Fig. 2b in Supplementary Note 2. A similar background of smaller particles than the sample particle size created by buffer balls were also detected in other aerosol experiments[31]. The fact that we are detecting the non-evaporative buffer residues using our laser scattering setup underlines the importance of detecting small particles using laser scattering while optimizing the injector and sample preparation conditions for SPI experiments, as those particles will be visible in X-ray scattering experiments and need to be sorted out before data analysis of the virus diffraction pattern analysis.

In Fig. 5b we show the scattering and the DMA-CPC size histograms for the ribosomes. The peak size of the DMA-CPC measurements at 20 nm matches well with the ribosome 70S size, however, this peak cannot be reproduced by the scattering data. The underlying particle size distribution for sizes >20 nm matches well, which may indicate that under vacuum conditions the 70S falls apart into the 30S and 50S subunits, which are too small to be detected in our scattering setup.

The smallest sample we measured the scattering off is ferritin. With the concentration and flow rates we used to generate ferritin aerosols, we create not only ferritin monomers but also clusters. This is confirmed in the DMA-CPC measurement on ferritin (see Fig. 5c, orange line). The highest peak is

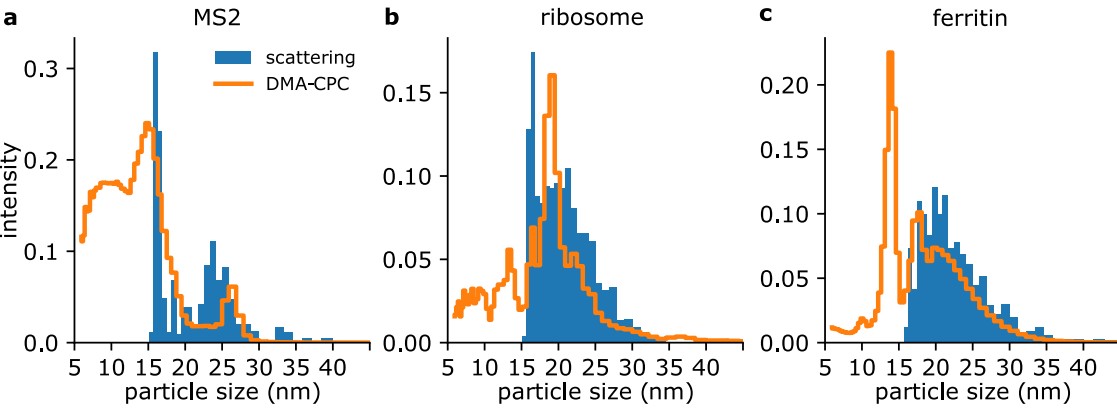

**Fig. 5 | Particle scattering results.** Comparison of particle size histograms measured with laser scattering (blue) and with a DMA-CPC (orange) for **a** MS2 virus particles, **b** ribosomes and **c** ferritin. The data is normalized to the area under the curve for particles >16 nm.

the ferritin monomer at a peak diameter of 13.5 nm, the dimer is present at 18 nm and larger clusters form the distribution >20 nm. In the scattering size histogram (Fig. 5c, blue), we are not able to detect ferritin monomers due to limitations in our imaging setup. In the size distribution from the scattering data we only observe particles in the size range of dimers and higher order clusters.

The scattering off MS2 and ferritin monomers may be approximated by the scattering of a sphere, but ribosomes and especially the clusters of ferritin are non-spherical particles. The diameter we are reconstructing is a light-scattering diameter, and not the actual diameter. By comparing the scattering diameter with the DMA-CPC diameter, we are comparing the light-scattering diameter to a hydrodynamic diameter, both diameters assume spherical particles and therefore, are expected to match reasonably well. In the size calculation of the bioparticles, we do not correct for the refractive index change. However because the reflective index is similar in this case, the change between PS and proteins can be neglected. If other materials are used for scattering experiment, this change in refractive index has a non-negligible effect on the retrieved particle size.

This particle detection method is able to retrieve the size histogram of the injected particles and potentially even that of particle mixtures. We assume if the size distributions of the individual particles is narrow enough and the peak sizes are further apart than the width of their distribution, we can disentangle even mixtures of particles.

The lower detection limit is given by the camera, the laser power and the laser intensity fluctuations. We determined the detection limit at around 16 nm. Detection of sub-16 nm particles could be achieved if the laser fluctuation is reduced to reduce the background fluctuation that can cause false hits in the data analysis, using a higher numerical aperture objective to collect a larger scattering angle and even tighter laser focus to increase the incident scattering intensity on the particles. All these limitations are setup-specific and can be addressed in a redesign of the interaction region to position an objective closer to the particle beam and a new laser with higher power to be installed. An alternative method to sub-16 nm bioparticle detection, apart from optical scattering, might be more interesting to pursue.

Optical scattering as a particle detection method is compatible with SPI experiments as a non-destructive method of particle detection and measures the particle size histogram directly in the interaction region. Compared to the DMA particle selection, our sizing method is independent of the charge, an additional advantage over DMA-CPC measurements.

From the data we collected, we determined the particle beam width for the different particle species we injected from the found particle positions within the ROI. Due to the tighter focus a characterization of the whole particle beam is more difficult compared to other studies[32] and we only determined the particle beam width at a fixed position. The particle beam width for all particle species is shown in Supplementary Fig. 6.

In addition to the single laser pulse data, we recorded data in the double pulse mode to trace individual particles and determine their speed from the delay time between the laser pulses and the position difference on the camera frames. Found particle positions from injecting the MS2 sample are shown in Supplementary Fig. 7a. The retrieved speed for the different particle species is shown in Supplementary Fig. 7b. We measured higher speeds for smaller particles. For 20 nm PS sample, for example, we measure a mean particle speed of 101 m/s and for the large 50 nm PS particles, we measure a speed of 76 m/s. We could not determine a speed for the ferritin sample due to the increased background from two laser pulses in one camera frame.

## Conclusion

Our results demonstrate that we can detect and determine the size of small viruses and proteins in a vacuum down to 16 nm compared to the previous detection limit of 40 nm polystyrene spheres. We were also able to detect the non-evaporative buffer balls that are generated during the ESI process. This type of information is very important during sample preparation. Another noteworthy finding was the discrepancy in the MS2 particle size between the DMA and the scattering data. The smaller size derived from the scattering data may be attributed to additional drying or partial collapse of the viral capsid under vacuum conditions. Similarly, scattering data from ribosome particles indicate that under vacuum conditions the 70S falls apart. Our results underline the importance of diagnostics that occur in the same conditions (e.g., in vacuum) as the final experiments. We expect this method to complement the sample injection and preparation steps used in current SPI experimental plans. Furthermore, we envision this method may find applications in other fields such as particle deposition, combustion research and mass spectrometry.

## Data availability

The data that support the findings of this study are available from the corresponding author upon reasonable request.

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

## Acknowledgements

We would like to thank Anna Munke and Diogo de Matos Filipe for preparing and providing the MS2 sample and buffer. We also would like to thank Suparna Sanyal's group for preparing and providing the ribosome sample. This work is supported by the Swedish Research Council (2018-00234 and 2019-06092) the Carl Tryggers Stiftelse för Vetenskaplig Forskning (CTS 19-227) and the European Research Council (ERC Consolidator Grant 101088426).

## Author contributions

T.Ye., L.W., and F.R.N.C.M. conceived the idea. T.Ye. developed and built the scattering setup. T.Ye. and L.W. carried out the experiments. T.Yo. wrote the hit-finding scripts and L.W. analyzed the results. The manuscript was written with input from all authors.

## Funding

## Competing interests

The authors declare no competing interests.
