## [Transparent Peer Review file · Communications Physics]

Aerosol size determination via light scattering of viruses and protein complexes

Corresponding Author: Professor Filipe Maia

Version 0:

Reviewer comments:

Reviewer #1

(Remarks to the Author)

In their manuscript, the authors present an optical apparatus for imaging of ultrafine particle aerosols. The presented method is a new implementation of their previously published work, that now allows the detection and size quantification of particles smaller than the previous detection limit of 40 nm. The new implementation allows the characterisation of aerosols containing biological particles such as viruses and large protein complexes. Specifically, the authors focus on integrating their method with single-particle X-ray diffractive imaging for better sample characterization under identical conditions. Their findings show the ability to measure monodisperse aerosols containing polystyrene particles at different diameters between 23 and 51 nm. They then use the linear relation between the sixth root of the scattering intensity and the particle diameter (as given by Rayleigh law) to generate a calibration curve. They further use the measured linear relation to measure the size distributions of three biological samples: Bacteriophage MS2, 70S ribosome and apo-ferritin complex.

Overall, the presented results are largely clear, showing improvement of the detection limit owing to the higher laser power density and the higher numerical aperture of the objective (however the exact value of the new detection limit is not clear). This allowed successful detection and quantification of MS2 bacteriophages (24 nm) and, to a lesser extent, of 70S ribosome (20 nm) and ferritin (13.5nm). Although I am not an expert in single-particle X-ray diffractive imaging, I found this method an important addition to the measurement setup allowing a rapid and straightforward characterisation of the delivered sample at the same conditions as the X-ray imaging measurement. In addition, for the case of large complexes such as viruses the results are convincing. The paper is probably suitable for publication, however, I do think that the manuscript would strongly benefit from some clarification which I summarized in the following points:

1. I have a concern regarding the writing, specifically the title and the abstract and their representation of the actual experimental results. The authors have named their work 'Aerosol Mass Photometry of Viruses and Protein Complexes', however, throughout the manuscript they show no evidence for their ability to actually provide a quantitative measure of the mass of the detected particles. Moreover, the word 'mass' is not mentioned in any of their results. What the authors demonstrated is the ability to detect the scattering intensity of the examined particles and, to a certain extent, to infer their size upon calibration with polystyrene particles. The authors draw an analogy with the solution method Mass Photometry, however, the fundamental principle that enables the supersensitive detection and quantification of proteins in solution, is the interferometric nature of the method combined with the manipulation of the reference and scattered fields. In contrast, the detection and sizing of larger particles (or stronger scatterers than proteins) were shown with 'dark-field' light scattering microscopy, which is more relevant to the approach presented here. Since the author only address the detection and the size of the particles and not their mass, I would suggest to modify the title to better reflect the actual results presented.

Additionally, the authors state in the abstract:

'Here we show the possibility to apply a similar principle to detect and measure the size and location of single viruses and protein complexes forming an aerosol beam, as well as trace their path.'

It was shown in their previous paper that the approach can be used to trace the path of individual particles, however in this work they show no data regarding tracking of individual particles. Here, the authors modified the optical apparatus to improve the detection limit by using tighter focus and as a result smaller field of view. It is not clear if it is possible to robustly track individual protein complexes given the field of view of the current setup and the lower SNR of smaller protein scatterers.

In the end of the introduction (line 96): 'This method can also be used as a general aerosol sizing technique when standard approaches are impractical or impossible'.

It is not clear how well this method can be applied when a mixture of particle in an aerosol (or inhomogeneous samples) are considered?

2. The authors show that with the current apparatus they were able to improve the detection limit of their optical approach. However, I find the method and results sections missing some important quantifications:

In my opinion, a quantification of the SNR as a function of the particles size would be extremely informative with respect to the detection limit, as well as quantification of the noise floor. What is the measured SNR vs. size and what is expected for a 15 nm particle in the current setup?

Regarding their calibration method with polystyrene particles, could the authors include some raw/denoised images to show the actual images of the point spread functions above the noise level for the different particle sizes, all on the same scale? Additionally, a visualisation of their method to eliminate the out of focus particles would also be helpful.

It is not entirely clear from the text why the threshold in the detection step needs to be a function of the size of the particles measured. Since all the particles are <50nm, where do the stray reflections come from? Did the authors use a varying threshold also for the protein samples?

As most sample do not consist of individual molecular species, as shown for the biological samples, an important question arises, can this method quantitatively detect and separate a solution containing mixtures of particles? and if yes, what is the achievable resolution. To address this point, could the author please add the size histograms of the detected PS particles (similar to Figure S2)? This will allow for a better comparison of the optical measurement with DMA-CPC and help the reader to better understand the measurement resolution.

3. Line 229: Since the scattering power is a function of the particle/molecular polarizability, to what extent do the authors expect the calibration with PS particle to be accurate for proteins? Could the difference in the diameter of the virus particle result from the different polarizabilities of the proteins and DNA compared with PS?

4. For the biological samples, the only resolved histogram is the measurement of the MS2 with a measured diameter of ~24 nm. For both the ribosome and ferritin, the results were inconclusive regarding the ability to detect/resolve the 20nm particles (ribosome) or the 13.5 and 18 nm particles of the ferritin, while both were resolved using DMA-CPC. While their explanation regarding the possible disassembly of the ribosome is plausible, it is also possible that these particles are actually below the detection limit for protein samples. How was the detection limit of 15 nm determined? It is clear from the histogram that a sharp cutoff <15 nm exists, but are the detected particles between 15 and 20 nm actually protein particles or are these artifacts from the buffer or false positive detection events? The authors suggest that these could come from buffer impurities or non-evaporating residues as well as laser fluctuations. In this regard, the authors should add histograms of the relevant buffer measurements without the proteins, measured and analyzed with the same parameters, to show that the detected particles near the limit of detection are proteins.

Some minor issues:

In the supporting information the authors state 'Ribosomes and MS2 were prepared in-house.' Please add the materials and methods relevant for the sample preparations or a relevant citation.

Table 1: For the ferritin, ribosome and bacteriophage, how did the authors determine the mean size (second column)?

Line: 236: The authors speculate that the sample purity was not high, which could result in the large peak at small sizes. Did the authors perform standard quantification of the solution distribution before their measurements (such as SEC or solution light scattering) to have a reference of the solution distribution?

'Detection of sub-15 nm particles could be achieved if the laser fluctuation is reduced to reduce the background fluctuation that can cause false hits in the data analysis, using a higher numerical aperture objective to collect a larger scattering angle and even tighter laser focus to increase the incident scattering intensity on the particles.'

The suggested modifications to the optical setup are similar to the modifications shown here compared to their previous work. Are there also limitations in going to higher NA, higher laser power and tighter focus? The authors should discuss.

Reviewer #2

(Remarks to the Author)

This paper improves upon mass/volume photometry for aerosol particles flowing out of an aerodynamic lens stack. The technique is useful for alignment and diagnostics of small particles in free space especially for experiments at X-ray Free Electron Lasers. The authors use a highly focused Nd:YAG laser with a 10x microscope system to image particles using Rayleigh scattering (e.g. correlating the intensity of the scattering to the size of the particles to the 6th power). It is very impressive that they are able to observe particles down to <20 nm in size and believe they can accurately measure particle sizes to ~15 nm with the setup. This all assumes the particles are spherical and that the calibration intensity from the slope for the polystyrene spheres (figure 3) really is applicable to all particle competitions.

The paper is well written and makes a good progress in capabilities of aerosol imaging.

There are few things in the paper that worry me. The first is that the description and analysis of figure 4 for in the text does not match the figure (e.g. dashed red line is not in the figure, nor are black lines). The histogram from the data appears to be a bar graph with a line graph of the DMA-CPC (which is an orange-ish line that could be considered red but is not dashed). I would encourage the authors to double check that the figure is the one they intended.

Figure 4 also worries me due to the scaling of the Y axis. Why was 25 nm the point chosen to make the DMA-CPC equal to the scattering data? Why not scale area under the curve >15 nm? Or some other normalization.

I am also a bit confused with the differences in sizes used in figure 3 with the supplemental figure 1a. The authors use the manufacture size distributions for the polystyrene spheres particles. It looks like either the DMA-CPC is off (calibration error/selection bias?) for the 50 nm particles for the the manufacture specifications which are used to calibrate the data (e.g. the 50 nm PS point is at 50 nm in figure 3 but in the DMA-CPC the data appares to show that the particles are >60 nm).

Lastly it would be nice to have in the supplemental frames of scattering intensities shown for the 20 nm and 50 nm polystyrene spheres particles as well as highlight the 25px and 5 px windows used for analysis. It would also help those who are unfamiliar with the Hamamatsu Orca Flash 4.0 V2 camera (sensitivity, readout noise, dynamic range, etc.) and explain the 15 nm resolution limit of the system.

Reviewer #3

(Remarks to the Author)

In the manuscript titled "Aerosol mass photometry of viruses and protein complexes", the authors size aerosol particles based on the amount of light scattered. Specifically, the authors apply this approach to size viruses, proteins and polystyrene nanoparticles present in an aerosol beam. The scope of the work is focussed on extending the principles of using light scattering to size particles for downstream single particle x-ray diffractive imaging applications. To put this work into a larger context of mass photometry, the authors main contribution is to apply this to aerosol particles rather than nanoparticles in aqueous medium. The authors provide proof of validation comparing the sizing with established techniques such as DMA-CPC.

The manuscript presents an interesting application for particle sizing and of high relevance and impact.. Nevertheless, given that the manuscript describes a technical method, it is this referee's opinion that the description of the methods, formulation of the measured images, characterisation of the platform, presentation of the critical image processing steps, interpretation of the data and some of the conclusions stated in the manuscript should be improved. As such the authors should address the concerns listed below before this work is considered for publication.

Major:

1. For mass photometry the signal arises from the interference between light scattered by the particle and a reference electric given by the weak reflection from the change in refractive index at the interface. In this manuscript, the light scattered by the incident beam corresponds to that of the aerosol containing the nanoparticle of interest. This begs the question whether the main signal contribution comes from the aerosol, i.e. non evaporative buffer and not the nanoparticle themselves. Furthermore, the authors should be articulate what control experiment they performed to ensure that what they measure are indeed single particles and monodisperse aerosols.
2. No images are shown of the data. The authors claim an imaging technique and perform a series of image processing steps but none of these are shown as data. The reader is only shown the final size distribution.
3. The size calibration is performed using particles with a known refractive index, which does not match that of the biological particles investigated. For absolute sizing based on light scattered by the particles the effective refractive index of the particle and that of the media should be considered.
4. The authors should better explain what the non-evaporative "buffer balls" refer to. Is there a difference between these entities and the aerosols containing particles.
5. The properties of the aerosol beam should be better characterised. Especially for a non-specialist in this field, it would be of great value to show the performance and capabilities of these aerosol beams.

Minor:

1. Half of the figures in the manuscript correspond to schematic diagrams.
2. The title claims that there is a mass measurement, "mass photometry", nevertheless the experiment only retrieves size, so this is quite misleading.

Version 1:

Reviewer comments:

Reviewer #1

(Remarks to the Author)

The authors have thoroughly addressed my comments regarding the presentation of their method. They have also added more technical information to allow better understanding of their analysis. I have no further comments and I find the manuscript suitable for publication.

Reviewer #2

(Remarks to the Author)

I think the authors addressed all of my previous comments and concerns. I am very happy with the updated manuscript.

Reviewer #3

(Remarks to the Author)

The authors have addressed all my concerns, and I am satisfied with the changes made to the manuscript. The authors' revision has improved the manuscript in terms of readability, clarity and the description of the methods. In my view, the manuscript is now ready for publication.

Reply to the reviewers

Reviewers' comments in blue, our comments/answers in black.

We thank the reviewers for their valuable feedback on our manuscript. We have made significant changes that the reviewers suggested and we address these in detail below:

Reviewer #1 (Remarks to the Author):

In their manuscript, the authors present an optical apparatus for imaging of ultrafine particle aerosols. The presented method is a new implementation of their previously published work, that now allows the detection and size quantification of particles smaller than the previous detection limit of 40 nm. The new implementation allows the characterisation of aerosols containing biological particles such as viruses and large protein complexes. Specifically, the authors focus on integrating their method with single-particle X-ray diffractive imaging for better sample characterization under identical conditions. Their findings show the ability to measure monodisperse aerosols containing polystyrene particles at different diameters between 23 and 51 nm. They then use the linear relation between the sixth root of the scattering intensity and the particle diameter (as given by Rayleigh law) to generate a calibration curve. They further use the measured linear relation to measure the size distributions of three biological samples: Bacteriophage MS2, 70S ribosome and apo-ferritin complex.

Overall, the presented results are largely clear, showing improvement of the detection limit owing to the higher laser power density and the higher numerical aperture of the objective (however the exact value of the new detection limit is not clear). This allowed successful detection and quantification of MS2 bacteriophages (24 nm) and, to a lesser extent, of 70S ribosome (20 nm) and ferritin (13.5nm). Although I am not an expert in single-particle X-ray diffractive imaging, I found this method an important addition to the measurement setup allowing a rapid and straightforward characterisation of the delivered sample at the same conditions as the X-ray imaging measurement. In addition, for the case of large complexes such as viruses the results are convincing. The paper is probably suitable for publication, however, I do think that the manuscript would strongly benefit from some clarification which I summarized in the following points:

1. I have a concern regarding the writing, specifically the title and the abstract and their representation of the actual experimental results. The authors have named their work 'Aerosol Mass Photometry of Viruses and Protein Complexes', however, throughout the manuscript they show no evidence for their ability to actually provide a quantitative measure of the mass of the detected particles. Moreover, the word 'mass' is not mentioned in any of their results. What the authors demonstrated is the ability to detect the scattering intensity of the examined particles and, to a certain extent, to infer their size upon calibration with polystyrene particles. The authors draw an analogy with the solution method Mass Photometry, however, the fundamental principle that enables the supersensitive detection and quantification of proteins in solution, is the interferometric nature of the method combined with the manipulation of the reference and scattered fields. In contrast, the

detection and sizing of larger particles (or stronger scatterers than proteins) were shown with 'dark-field' light scattering microscopy, which is more relevant to the approach presented here. Since the author only address the detection and the size of the particles and not their mass, I would suggest to modify the title to better reflect the actual results presented.

We agree with you that we do not show a measurement of the mass in this manuscript. We have modified the title of the paper to "Aerosol Size determination via light scattering of viruses and protein complexes" to reflect its content. With this title, we ensure that there is no confusion in the measurement procedure with mass photometry measurements. We also modified the abstract by removing the comparison with the mass photometry principle to avoid confusion and focus on the measurement of the size of the particles.

Additionally, the authors state in the abstract:

'Here we show the possibility to apply a similar principle to detect and measure the size and location of single viruses and protein complexes forming an aerosol beam, as well as trace their path.'

It was shown in their previous paper that the approach can be used to trace the path of individual particles, however in this work they show no data regarding tracking of individual particles. Here, the authors modified the optical apparatus to improve the detection limit by using tighter focus and as a result smaller field of view. It is not clear if it is possible to robustly track individual protein complexes given the field of view of the current setup and the lower SNR of smaller protein scatterers.

It is possible to track individual particles with the modified setup, which we included in the updated manuscript. To track individual particles, we performed additional measurements with the double-pulse mode of the laser. We included the results in the main manuscript and added a figure in the supplementary material (Figure S7). Fig. S7a shows the particle positions determined for MS2 particle double hits and Fig. S7b shows the particle speed determined for the different particle species. We are able to track individual particles of all samples, except from the ferritin sample due to a higher background (generated from the now two laser pulses).

We added the following paragraphs in the main manuscript, section 2.2 ll. 198ff.:

"To trace individual particles and determine the particle's speed, we recorded frames with the laser in double-pulse mode. The delay of the pulses was set to 0.3 μ s for the PS sample and to 0.2 μ s for the biosamples. Each frame corresponds to a double laser shot. The data analysis was performed similar to the data analysis of the single pulse data with the extension of finding frames with a double hit and determining the difference in distance between the hits to determine the particle's speed."

And in ll. 382 ff.:

"In addition to the single laser pulse data, we recorded data in the double pulse mode to trace individual particles and determine their speed from the delay time between the laser pulses and the position difference on the camera frames. Found particle positions from injecting the MS2 sample are shown in the supplementary material Fig. 7a. The retrieved speed for the different particle species is shown in the supplementary material Fig. 7b. We measured higher speeds for smaller particles. For 20 nm PS sample, for example, we measure a mean particle speed of 101 m/s and for the large 50 nm PS particles, we measure a speed of 76 m/s. We could not determine a speed for the ferritin sample due to the increased background from two laser pulses in one camera frame."

In the end of the introduction (line 96): 'This method can also be used as a general aerosol sizing technique when standard approaches are impractical or impossible'.

It is not clear how well this method can be applied when a mixture of particle in an aerosol (or inhomogeneous samples) are considered?

For this manuscript, we did not take data on mixtures of samples. Because of the broad size distributions of the PS sample (observed in the DMA and in optical scattering histograms), we assume a distinction between e.g., the 20 and 30 nm PS peaks is not possible.

We assume then that if the size distribution is narrow (e.g. as for the MS2), and the peaks are further apart than the width of their distribution, we can resolve it in the optical scattering, too.

We have added a statement about the possibility of measuring mixtures in the discussion part (ll. 340 ff.):

"This particle detection method is able to retrieve the size histogram of the injected particles and potentially even that of particle mixtures. We assume if the size histogram of the individual particles is narrow enough and the peak sizes are further apart than the width of their distribution, we can disentangle even mixtures of particles."

2. The authors show that with the current apparatus they were able to improve the detection limit of their optical approach. However, I find the method and results sections missing some important quantifications:

In my opinion, a quantification of the SNR as a function of the particles size would be extremely informative with respect to the detection limit, as well as quantification of the noise floor. What is the measured SNR vs. size and what is expected for a 15 nm particle in the current setup?

Regarding their calibration method with polystyrene particles, could the authors include some raw/denoised images to show the actual images of the point spread functions above the noise level for the different particle sizes, all on the same scale? Additionally, a visualisation of their method to eliminate the out of focus particles would also be helpful.

This useful information was indeed missing. We have added the results for the signal strength depending on the particle size in the results section and the noise levels to complete our data presentation. The noise level was determined on background images taking into account the shot-to-shot pixel intensity differences. Two figures were added to the manuscript (Fig. 4a and b) that show our determination of the 16 nm particle detection limit. The limit is determined with the PS diameters as peak diameters from the DMA data, not taking the manufacturers' size due to their deviation as pointed out by Reviewer #2.

We added the following to the manuscript (lines 230ff.):

"We determined the lower detection limit in our setup to be 16 nm. This value is a result of the shot-to-shot pixel intensity difference (noise) and the background level (background). The noise distribution is shown in Figure 4a. The dashed black line shows the average distribution. In Figure 4b, we show the PS hit intensity to background ratio (blue) with a sixth-root fit through the data (orange line). For the large 59 nm PS particles, the signal is 240 times larger than the background and this ratio decreases to 3.3 for the 18 nm PS particles. The lower detection limit is determined by the noise from shot to shot. This is represented by the gray line. The

value is taken as the $1/e^2$ value of the distribution in Figure 4a. The two lines cross at a particle size of 16 nm, which we set as our lower detection limit. For this size, the intensity would only be 2.8 times larger than the background.”

In addition, we added a raw and denoised data frame containing a particle hit from a measurement when injecting ferritin in the main manuscript (Figure 2). This figure also contains a visualization of the two circular windows to determine the focused particles and an unfocused particle hit as an example. We refer to this figure in ll. 168 ff.:

“An example of a single raw frame containing a particle hit is shown in Figure 2a. The raw data frames were corrected for dead pixels on the camera (see center of frame in Figure 2a) and the mean laser background, which was recorded without sample being delivered, was subtracted as shown in Figure 2b.”

And in ll. 193 ff.:

“Examples of a focused and an unfocused particle hit are shown in Figure 2c and Figure 2d, respectively.”

We have also added exemplary focused hits of all used PS sizes and bioparticles in the supplementary material for comparison on the same scale as suggested (supplementary Figure 4), which we refer to from the main manuscript in the caption of Figure 4:

“Focused particle examples for all particle sizes used are shown in the supplementary material Figure 4.”

It is not entirely clear from the text why the threshold in the detection step needs to be a function of the size of the particles measured. Since all the particles are <50nm, where do the stray reflections come from? Did the authors use a varying threshold also for the protein samples?

We analysed the data again with a fixed threshold to avoid this additional parameter in the data analysis. Initially, we used the varying threshold because we started the analysis with large particles (larger than presented here) and when going to smaller particles, the high threshold excluded particles that we wanted to detect. A high threshold for the larger particles was used to filter clearly unfocused and therefore weak hits from the analysis early. We later added an analysis step that determines whether a particle is focused or not based on the area the particle hit occupies, which removes these unwanted hits. We now decided to use a fixed threshold for all particle sizes including the protein samples.

We removed the sentence about the varying threshold from the methods and added the value we used as a fixed threshold in lines 177ff.:

“We then threshold the denoised image to determine the pixels with photons with a fixed threshold of 20.”

As most sample do not consist of individual molecular species, as shown for the biological samples, an important question arises, can this method quantitatively detect and separate a solution containing mixtures of particles? and if yes, what is the achievable resolution. To address this point, could the author please add the size histograms of the detected PS particles (similar to Figure S2)? This will allow for a better comparison of the optical measurement with DMA-CPC and help the reader to better understand the measurement resolution.

Yes, our method can in principle detect and separate a solution containing differently sized particles. However, we point out that in order to separate two particle sizes, their retrieved size distribution has to be narrow enough and the peak sizes further apart than the width of their distributions. As it is visible in the added DMA and scattering data for the PS (supplementary figure 5), this would be difficult for e.g. 20 and 30 nm PS in a mixture.

We have added the size histogram of the detected PS particles in the supplementary material Figure 5 and refer to the figure in the main manuscript in lines 225 ff.:

“The retrieved size histograms calculated from the scattering data of the PS samples based on the calibration is shown in the supplementary material Fig. 5 in comparison to the DMA-CPC size histogram. “

3. Line 229: Since the scattering power is a function of the particle/molecular polarizability, to what extent do the authors expect the calibration with PS particle to be accurate for proteins? Could the difference in the diameter of the virus particle result from the different polarizabilities of the proteins and DNA compared with PS?

The scattering intensity depends on the refractive index of the particle. The calibration with PS particles is accurate, as long as the measured particle has a similar refractive index. At the used laser wavelength of 532 nm, the refractive index of PS is 1.599 [Karsarova *et al.*, *Optical Materials* **29**, 11, 1481 (2007)]. An average refractive index for human proteins is 1.60346 and for E.coli 1.60123. Neglecting the imaginary part, a change in refractive index from PS to protein will result at the same scattering intensity in a size with a factor 0.998 for human proteins and 0.999 for E.coli difference. At the MS2 virus size, this means a shift of less than 0.3 nm, which can be neglected here.

We speculate the smaller MS2 size is a result of collapsing of the virus in vacuum conditions, similar to what Mall *et. al.* observed in an SPI experiment [Mall *et. al.*, *submitted*, arXiv:2407.11687 [q-bio.BM]]. The reference was added in the main manuscript to line 267. If other materials are used in the scattering setup that have a very different refractive index, we would have to account for that refractive index change and sizes would appear larger/smaller depending on the refractive index being smaller/larger.

We have added the following statement in the results section of the manuscript, line 332ff.:

“In the size calculation of the bioparticles, we do not correct for the refractive index change. However because the refractive index is similar in this case, the change between PS and proteins can be neglected. If other materials are used for scattering experiments, this change in refractive index has a non-negligible effect on the retrieved particle size.”

4. For the biological samples, the only resolved histogram is the measurement of the MS2 with a measured diameter of ~24 nm. For both the ribosome and ferritin, the results were inconclusive regarding the ability to detect/resolve the 20nm particles (ribosome) or the 13.5 and 18 nm particles of the ferritin, while both were resolved using DMA-CPC. While their explanation regarding the possible disassembly of the ribosome is plausible, it is also possible that these particles are actually below the detection limit for protein samples. How was the detection limit of 15 nm determined? It is clear from the histogram that a sharp cutoff

<15 nm exists, but are the detected particles between 15 and 20 nm actually protein particles or are these artifacts from the buffer or false positive detection events? The authors suggest that these could come from buffer impurities or non-evaporating residues as well as laser fluctuations. In this regard, the authors should add histograms of the relevant buffer measurements without the proteins, measured and analyzed with the same parameters, to show that the detected particles near the limit of detection are proteins.

To say with confidence that the particles near the detection limit are actual particles, we performed two additional experiments: We focused on the MS2 sample in this case:

First, we injected a 20mM AmAc solution and collected data frames. In our analysis, no particles were found. This excludes laser fluctuations or tiny aerosol particles and the gas background being detected as false positive hits.

Second, we injected the MS2 buffer, which is a Tris-buffer, without any virus particles in it. We collected data frames and analysed them. We detected particles in the size range corresponding to the DMA-CPC size histogram of the Tris-buffer.

We added these results in the supplementary material and added the relevant size histogram in the supplementary material Figure 2b.

We refer to this finding and extent the description of the small particles in the main text in ll. 275 ff.:

“Initial MS2-empty droplets in the electrospray process can contain aggregates of the buffer residues and create spheres in this size range that we refer to as buffer balls. To support this idea, we have recorded data while injecting the MS2 buffer without MS2 virus particles. We observed particle hits with small diameters only. This data is presented in the supplementary material Figure 2b. A similar background of smaller particles than the sample size created by buffer balls were also detected in other aerosol experiments [31].”

[31] Bielecki et al., Sci. Adv. 5, eaav8801 (2019)

Some minor issues:

In the supporting information the authors state ‘Ribosomes and MS2 were prepared in-house.’ Please add the materials and methods relevant for the sample preparations or a relevant citation.

We have changed the statement from

“Ribosomes and MS2 were prepared in-house.”

in the supplementary material to:

“Ribosomes were prepared in-house according to the protocol given in [1]. The MS2 sample was prepared following a modified protocol based on [2]. The virus was propagated using E. coli strain K12 (ATCC 10798). After the precipitation steps and re-suspension in Tris buffer, the sample was further purified on a Sepharose CL-4B (Cytiva) column (500 mL). Peak fractions were pooled and precipitated overnight at 4°C using 10 % PEG 6000 and 0.5 M NaCl (from a 5 M NaCl solution). The precipitate was then centrifuged at 27000xg for 30 min, and the resulting pellet was re-suspended in Tris buffer. Prior to injection, the Tris buffer was exchanged to 20 mM AmAc using aPD Minitrap G-25 column (Cytiva). .”

1. Johansson *et al.*, *Molecular Cell* **30**, 5, 589-598 (2008)
2. Mall *et al.*, arXiv:2407.11687v1 [q-bio.BM] (2024)

Table 1: For the ferritin, ribosome and bacteriophage, how did the authors determine the mean size (second column)?

The sizes were taken from the literature, which we added in the supplementary material.

Line: 236: The authors speculate that the sample purity was not high, which could result in the large peak at small sizes. Did the authors perform standard quantification of the solution distribution before their measurements (such as SEC or solution light scattering) to have a reference of the solution distribution?

We did not perform such measurements to determine the solution distribution. Typically, we determine the sample purity in our experiments from the DMA-CPC size histogram. If the histogram shows a clear and isolated sample peak in the expected size range, we assume it is clean enough for our experiments. The DMA-CPC measurement is a size measurement on the aerosol droplets and comparable to the SPI experiments we are aiming for. The sample quality in SEC or dynamic light scattering would not give us information about the droplet composition of the aerosol and is not used in preparation and for comparison of the retrieved size histogram from optical scattering.

'Detection of sub-15 nm particles could be achieved if the laser fluctuation is reduced to reduce the background fluctuation that can cause false hits in the data analysis, using a higher numerical aperture objective to collect a larger scattering angle and even tighter laser focus to increase the incident scattering intensity on the particles.'

The suggested modifications to the optical setup are similar to the modifications shown here compared to their previous work. Are there also limitations in going to higher NA, higher laser power and tighter focus? The authors should discuss.

All the limitations we mention are specific limits in our setup and could be mitigated with a redesign of the whole setup: The laser power we use is the maximum provided by the used laser. A tighter laser focus is an option to increase the scattering intensity and will be explored in future experimental campaigns. A higher NA objective is currently not feasible due to physical limitations of working distance and space inside the vacuum chamber, i.e. a higher NA objective with a shorter working distance would hit the aerodynamic lens injector. A redesign of the outside geometry of the injector could avoid this.

We have added a statement in the discussion section of the manuscript as follows in line 357ff.:

"All these limitations are setup-specific and can be addressed in a redesign of the interaction region to position an objective closer to the particle beam and a new laser with higher power to be installed. An alternative method to sub-16 nm bioparticle detection, apart from optical scattering, might be more interesting to pursue."

Reviewer #2 (Remarks to the Author):

This paper improves upon mass/volume photometry for aerosol particles flowing out of an aerodynamic lens stack. The technique is useful for alignment and diagnostics of small

particles in free space especially for experiments at X-ray Free Electron Lasers. The authors use a highly focused Nd:YAG laser with a 10x microscope system to image particles using Rayleigh scattering (e.g. correlating the intensity of the scattering to the size of the particles to the 6th power) . It is very impressive that they are able to observe particles down to <20 nm in size and believe they can accurately measure particle sizes to ~15 nm with the setup. This all assumes the particles are spherical and that the calibration intensity from the slope for the polystyrene spheres (figure 3) really is applicable to all particle competitions.

The paper is well written and makes a good progress in capabilities of aerosol imaging.

There are a few things in the paper that worry me. The first is that the description and analysis of figure 4 for in the text does not match the figure (e.g. dashed red line is not in the figure, nor are black lines). The histogram from the data appears to be a bar graph with a line graph of the DMA-CPC (which is an orange-ish line that could be considered red but is not dashed). I would encourage the authors to double check that the figure is the one they intended.

Thank you for pointing out the mistake, we had forgotten to update the text when we changed the figures. We have corrected the error.

Figure 4 also worries me due to the scaling of the Y axis. Why was 25 nm the point chosen to make the DMA-CPC equal to the scattering data? Why not scale area under the curve >15 nm? Or some other normalization.

We chose to scale the DMA data to the scattering data at 25 nm, because we were confident that we would detect 25 nm particles and larger from the beginning, which was supported by the size distribution falloff matching nicely towards larger particles. However, to avoid confusion and bias in our data analysis, we changed the normalization to the area under the curve >16 nm (detection limit, see next comment).

I am also a bit confused with the differences in sizes used in figure 3 with the supplemental figure 1a. The authors use the manufacture size distributions for the polystyrene spheres particles. It looks like either the DMA-CPC is off (calibration error/selection bias?) for the 50 nm particles for the the manufacture specifications which are used to calibrate the data (e.g. the 50 nm PS point is at 50 nm in figure 3 but in the DMA-CPC the data appears to show that the particles are >60 nm).

We measure the 50 nm PS sample repeatedly at a peak size of 59 nm in the DMA at different settings of the DMA. A DLS measurement of the 50 nm PS sample supports the DMA peak size and returns a mean diameter of 58.5 nm. We do not know why the manufacturer's size is so different. A statement about the different sizes is added in the supplementary material in the sample details section.

In addition, we have updated the calibration of our setup using the peak diameter we obtain from the DMA-CPC measurements instead of using the manufacturers' provided particle size for the PS samples. This change had an impact on the calibration curve of course, which changed in the slope and overall the results of the retrieved sizes from the scattering analysis of the bioparticles. We have changed this throughout the Results and Discussion section. Our new detection limit is 16 nm, which is explained in the main manuscript in more detail now.

Lastly it would be nice to have in the supplemental frames of scattering intensities shown for the 20 nm and 50 nm polystyrene spheres particles as well as highlight the 25px and 5 px windows used for analysis. It would also help those who are unfamiliar with the Hamamatsu Orca Flash 4.0 V2 camera (sensitivity, readout noise, dynamic range, etc.) and explain the 15 nm resolution limit of the system.

We have included a new figure in the main manuscript (Figure 2) showing a raw and denoised frame (Figure 2a and b) with a particle hit and includes a visualization of the circular windows (Figure 2c and d) to determine whether the particle is focused or not. We refer to the figure in the main text in ll. 168ff.:

“An example of a single raw frame containing a particle hit is shown in Figure 2a. The raw data frames were corrected for dead pixels on the camera (see center of frame in Figure 2a) and the mean laser background, which was recorded without sample being delivered, was subtracted as shown in Figure 2b.”

And in ll. 193ff.:

“Examples of a focused and an unfocused particle hit are shown in Figure 2c and 2d, respectively.”

We also added details on the resolution limit of the system and added the following paragraph to the main manuscript in ll. 247ff. And a new figure (Figure 4):

“We determined the lower detection limit in our setup to be 16 nm. This value is a result of the shot-to-shot pixel intensity difference (noise) and the background level (background). The noise distribution is shown in Figure 4a. The dashed black line shows the average distribution. In Figure 4b., we show the PS hit intensity to background ratio (blue) with a sixth-root fit through the data (orange line). For the large 59 nm PS particles, the signal is 240 times larger than the background. This ratio decreases to 3.3 for the 18 nm PS particles. The lower detection limit is determined by the shot-to-shot noise. This is represented by the gray line. The value is taken as the $1/e^2$ value of the distribution in Figure 4a. The two lines cross at a particle size of 16 nm, which we set as our lower detection limit. For this size, the intensity would only be 2.8 times larger than the background.”

Reviewer #3 (Remarks to the Author):

In the manuscript titled “Aerosol mass photometry of viruses and protein complexes”, the authors size aerosol particles based on the amount of light scattered. Specifically, the authors apply this approach to size viruses, proteins and polystyrene nanoparticles present in an aerosol beam. The scope of the work is focussed on extending the principles of using light scattering to size particles for downstream single particle x-ray diffractive imaging applications. To put this work into a larger context of mass photometry, the authors main contribution is to apply this to aerosol particles rather than nanoparticles in aqueous medium. The authors provide proof of validation comparing the sizing with established techniques such as DMA-CPC.

The manuscript presents an interesting application for particle sizing and of high relevance and impact.. Nevertheless, given that the manuscript describes a technical method, it is this

referee's opinion that the description of the methods, formulation of the measured images, characterisation of the platform, presentation of the critical image processing steps, interpretation of the data and some of the conclusions stated in the manuscript should be improved. As such the authors should address the concerns listed below before this work is considered for publication.

Major:

1. For mass photometry the signal arises from the interference between light scattered by the particle and a reference electric given by the weak reflection from the change in refractive index at the interface. In this manuscript, the light scattered by the incident beam corresponds to that of the aerosol containing the nanoparticle of interest. This begs the question whether the main signal contribution comes from the aerosol, i.e. non evaporative buffer and not the nanoparticle themselves. Furthermore, the authors should be articulate what control experiment they performed to ensure that what they measure are indeed single particles and monodisperse aerosols.

One control experiment to ensure the production of single particles and monodisperse aerosols is the DMA-CPC measurement. This measurement produces a size distribution of the generated aerosol, which matches the diameters of the particles according to the manufacturer or according to literature and shows us if we produce single particles or e.g. clusters and how much of the buffer is present in our generated aerosol in the form of aggregates. If, e.g., a non-evaporative buffer surrounds the particle of interest, the size in the DMA-CPC would appear larger than expected and the size would shift with the initial droplet diameter, which can be controlled via the flow rate of the liquid. In such cases, we repeat the measurement at a lower flow rate to generate an aerosol with a clean particle. In a previous experiment, it has been shown that the residue on electrosprayed samples can be minimized [Bielecki et al., Sci. Adv., 5, eaav8801 (2019)].

Even if the particle is not surrounded with a non-evaporative buffer, the aerosol itself can contain small non-evaporative "buffer balls", initially particle-empty droplets that do not evaporate completely. This cannot be completely avoided, but another indicator that we are measuring the particles in our scattering setup and not only the non-evaporative buffer background is that the measured scattering intensity scales to the power of 6 with the particle diameter.

The last checkpoint is a measurement of 20 mM AmAc solution without any particles inside. If we spray and inject this solution into our optical scattering experiment, we do not detect one single hit.

All of the above mentioned points combined are good indicators that we are indeed measuring single particles in our setup.

2. No images are shown of the data. The authors claim an imaging technique and perform a series of image processing steps but none of these are shown as data. The reader is only shown the final size distribution.

We have made significant changes to the manuscript and the supplementary material in line with comments from Reviewer 1 and 2 to include more figures that also show the actual data (new Figure 2, SI Figure 4) , the image processing steps (new Figure 2) and further particle

beam distributions (SI Figure 6). We hope these additions to the manuscript provide a better understanding of the image processing.

3. The size calibration is performed using particles with a known refractive index, which does not match that of the biological particles investigated. For absolute sizing based on light scattered by the particles the effective refractive index of the particle and that of the media should be considered.

If the refractive index is different from the calibration particles, the refractive index has to be considered. In our case, the calibration with PS particles is accurate, as long as the measured particle has a similar refractive index. At 532 nm, the refractive index of PS is 1.599 [Karsarova *et al.*, *Optical Materials* **29**, 11, 1481 (2007)]. An average refractive index for human proteins is 1.60346 and for E.coli 1.60123. Neglecting the imaginary part, a change in refractive index from PS to protein will result at the same scattering intensity in a size with a factor 0.998 for human proteins and 0.999 for E.coli. At the MS2 virus size, this means a shift of less than 0.3 nm, which can be neglected.

If other materials are used in the scattering setup that have a very different refractive index however, we agree to the statement, we would have to account for that refractive index change, otherwise, sizes would appear larger/smaller depending on the refractive index being smaller/larger.

We have added a statement in the results section of the manuscript, line 332ff.:

“In the size calculation of the bioparticles, we do not correct for the refractive index change. However because the reflective index is almost similar in this case, the change between PS and proteins can be neglected. If other materials are used for scattering experiments, this change in refractive index has a non-negligible effect on the retrieved particle size.”

4. The authors should better explain what the non-evaporative “buffer balls” refer to. Is there a difference between these entities and the aerosols containing particles.

The aerosol that is generated in the electrospray contains droplets with different content: 1) only solvent and 2) solvent and sample.

The “buffer balls” are created from the droplets containing only solvent. If the solvent is not exclusively a volatile buffer (such as AmAc), a solid residue remains of this droplet, creating a so-called buffer ball.

The second kind of droplets can generate isolated sample particles or particles covered in a small layer of non-volatile solvent content.

We have updated the manuscript to include a small description of the buffer balls in line 275 ff.:

“Initial MS2-empty droplets in the electrospray process can contain aggregates of the buffer residues and create spheres in this size range that we refer to as buffer balls. To support this idea, we have recorded control data while injecting the MS2 buffer without MS2 virus particles. We observed particle hits with small diameters only. This data is presented in the supplementary material in Figure 2b. A similar background of smaller particles than the sample particle size created by buffer balls were also detected in other aerosol experiments [31]. ”

[31] Bielecki *et al.*, *Sci Adv.* **5**, eaav8801(2019)

5. The properties of the aerosol beam should be better characterised. Especially for a non-specialist in this field, it would be of great value to show the performance and capabilities of these aerosol beams.

We have added the characterization of the particle beam width in the supplementary material for interested readers. For this study, we measured at a fixed distance from the injector exit, so we do not have extensive data on the particle beam evolution, but we show the shape and width of the particle beam at a fixed distance of 2 mm for the various samples in the supplementary material Figure 6. Compared to other studies on the particle-beam evolution, e.g. by Hankte et al. (Hankte *et al.*, *IUCrJ* **5**, 673–680 (2018)), our setup has more difficulties characterizing the whole particle beam due to the small laser focus size.

We are referring to the particle beam width measurements in the main manuscript in lines 372 ff.:

“From the data we collected, we determined the particle beam width for the different particle species we injected from the found particle positions within the ROI. Due to the tighter focus a characterization of the whole particle beam is more difficult compared to other studies [32] and we only determined the particle beam width at a focused position. The particle beam width for all particle species is shown in the supplementary material Figure 6. “

[32] Hankte *et al.*, *IUCrJ* **5**, 673–680 (2018)

Minor:

1. Half of the figures in the manuscript correspond to schematic diagrams.

With the overall changes we did to the manuscript, we now included significantly more figures including data and analysis than schematic diagrams. We also moved the second schematic diagram showing the overall setup (old Figure 2) to the supplementary material as Figure 3.

2. The title claims that there is a mass measurement, “mass photometry”, nevertheless the experiment only retrieves size, so this is quite misleading.

We have changed the title accordingly to: “Aerosol Size determination via light scattering of viruses and protein complexes”, because we indeed only retrieve the sizes of the particles.